# Software Development and Tool Support for Curriculum Design: A Systematic Mapping Study

Aliwen Melillán , Ania Cravero * and Samuel Sepúlveda

Software Engineering Studies Center, Computer Science and Informatics Department, Universidad de La Frontera, Temuco 4811230, Chile; a.melillan02@ufromail.cl (A.M.); samuel.sepulveda@ufrontera.cl (S.S.)
* Correspondence: ania.cravero@ufrontera.cl

**Abstract:** Curriculum design is the systematic process of establishing how a learning process is designed, developed, supported, and delivered. This process is supported by software tools which can help improve curriculum alignment and facilitate the design of courses or programs. This article aims to analyze software proposals for curriculum design support that consider using models, methods, and techniques in software development. To do this, a systematic mapping of studies was conducted, including six research questions. This study includes 45 articles published from 2011 to 2022. The results indicate that 60% use some model, method, or technique in software development. Most software uses some models, such as ontologies, UML diagrams, or IMS-1D models. Although most articles use some model, method, or technique, there is a lack of use of software engineering models such as UML diagrams, which are standard in the software industry and research.

**Keywords:** curriculum design; curriculum alignment; software development; tool support; systematic mapping study



## 1. Introduction

Higher education organizations have various programs of study that cover the area of undergraduate, postgraduate, and various training courses. These programs are designed through a process called "instructional design" or "curriculum design", which is established by the organization itself based on education regulations in the country of residence [1]. According to Dodd, curriculum design is a fundamental pillar of the way education takes place since at the core of this process is a mental model of how people learn, representing a design of how the transfer of knowledge and skills from theory to practice occurs [2].

In this sense, curriculum design is developed as a process that establishes activities such as planning, organizing, and integrating curricular components to achieve the expected learning outcomes [3]. Curriculum design is a systematic process that determines how the learning process is constructed, developed, supported, and delivered [3]. This process should consist of learning content, teaching methods, and evaluation methods to achieve the expected learning outcomes [4]. The curriculum design includes three levels: macro-curriculum, meso-curriculum, and micro-curriculum [5,6]. The macro-curriculum level considers curriculum planning at the country level. The meso-curriculum level considers curriculum planning at the institutional level. The micro-curriculum level considers course-level planning.

A recurring problem in this process is the alignment of curriculum components in several ways. The first is aligning programs at the meso-curriculum level with the needs of the industry [7] since it is necessary to create profiles and a set of competencies suitable for professional development. Second, a lack of alignment between micro-curriculum components, such as learning outcomes, methodologies, assessments, content, and indicators [8] should be avoided, as both outcomes and content must be aligned to ensure that the content

being presented supports students as they work to achieve specified learning outcomes [2]. Third, a lack of alignment between the components of curriculum design and the Information technologies (IT) that support this process [9,10] can represent a problem, as these are developed by a technical team that does not necessarily understand the meaning and relationships of these components [11].

This lack of alignment is due to the common understanding among the curriculum design development team. There is no common language between curriculum developers, teachers, students, company personnel, and the IT team [2,12]. In order to improve this aspect and others related to management, several software tools have been developed to support the process, either at the micro, meso, or macro-curriculum levels. However, more information is needed regarding the methods and models used for their development. To fulfill this need is important since the models represent the curricular elements and their relationships, making it possible to identify the crucial aspects to achieve alignment between IT and the curriculum design team. On the other hand, the methods must be aligned with the needs of the curriculum design team, which includes aspects of education, the characteristics of the curriculum, the needs of the market, and the characteristics of the educational model to be included, a dynamic and complex aspect to address [13].

Carvajal-Ortiz and Florian-Gaviria analyzed the models and software applications to support courses' design, evaluation, and analysis for competency-based curriculum management [14]. The authors showed that the software applications found only considered some of the characteristics that supported a competency-based curriculum, in addition to the alignment between the tool itself and the curriculum achieved.

On the other hand, Dodd analyzes various models used to support curriculum design. One example is "Canvas", a visual tool that can be used to plan and design a curriculum in a structured and coherent way on the same plane at a conceptual level through a brainstorming process [2]. Another example is the "Curriculum Matrix", which focuses on representing relationships and alignment between key variables in the curriculum. Despite publicizing the benefits of these models, software tools are not included to support the process. To include these tools is important since the software allows for streamlining processes, managing resources, and analyzing different possibilities, providing critical information to the work team through reports.

Modeling has been advocated for years as an essential part of software development to address complexity by providing abstractions and hiding technical details. Due to its wide application, numerous informal and formal approaches have been developed, such as Entity–Relationship Diagrams for modeling data, Specification and Description Language for modeling telecommunication systems, formal modeling languages such as Z and B, ontologies, and the Unified Modeling Language (UML), which is currently the most widely used in the industry [15].

Modeling was initially applied for stakeholder communication and to provide sketches (also called models or diagrams) of what a software system should do or its design. Today, the industry is increasingly using models for tasks other than system description, for example, simulation, test case generation, and parts or all of the source code, improving quality, lowering development costs, and increasing reliability [16].

On the other hand, the correct development of the models is carried out through methods and techniques from software engineering. These methods and techniques allow the models to be correct, complete, consistent, and understandable, increasing effective communication between stakeholders [17].

This study aims to identify the methods, models, and techniques used to develop curriculum design support tools. Since teachers are the leading educational software developers [18], this study may be relevant to identify the software models, methods, and techniques used for curriculum design specifically. For this purpose, we conducted a systematic mapping study according to the proposal of Petersen et al. [19]. Forty software tools that were used to support curriculum design were analyzed. In addition to identifying methods, models, and techniques, those that help with alignment were analyzed.

In addition, the main problems of software developers and curriculum developers when developing a software tool to support curriculum design were identified. Finally, a conceptual model is proposed that explains the relationships between the models, methods, and techniques used in the articles analyzed, allowing the generation of a knowledge base for interdisciplinary work among curriculum developers, program directors, teachers, and software developers.

Furthermore, this study can help develop new models to better represent the work team's needs and the relationships between the curricular components to achieve alignment. The impact of this study can be seen in several ways:

- To generate a knowledge base on how software tools for curriculum support are being developed;
- To understand the need for alignment between curriculum components;
- To generate new software models that enable alignment;
- The design of models that allow interdisciplinary work, such as meta-models, since they include rules for relating concepts.

This article is structured as follows. Section 2 describes the background. Section 3 specifies the methodology used. Section 4 shows the results obtained from the systematic mapping performed. Section 5 discusses the results obtained. Section 6 describes related work. Section 7 details the threats to the validity of the results together with mitigation techniques. Finally, Section 8 presents the conclusions and future work.

## 2. Background

This section describes the general concepts of curriculum design and software tools. In addition, the concepts of modeling, techniques, and development methods are explained.

### 2.1. Curriculum Design

This section describes the strategies for curriculum design and the curriculum levels.

#### 2.1.1. Curriculum Design Strategies

The two most common models for organizing pedagogical activities that give rise to curriculum design are the competency-based approach and the objectives-based approach [20,21]. The former is dominated by knowledge, its emotional resources, and how to insert them into the professional world. The second focuses on the efficient achievement of objectives from teaching theory. The competency-based approach has gained relevance in recent years, leading educational institutions and teachers to modify teaching and learning strategies to apply this approach. Competency-based curriculum design has specific characteristics [22]:

- It is designed based on the needs of society and not on knowledge that is passed down by tradition in educators and does not take into account the needs of society.
- It is centered on the student and their capabilities so that they can design feasible solutions as the central axis, and put the teacher as their guide [23].
- The evaluation focused on learning outcomes, i.e., how the student applies knowledge, allowing the student to evaluate him/herself and learn autonomously to apply knowledge successfully.
- Flexibility in learning opportunities, considering each student's own time to learn.

Competency-based curriculum design has different conceptions. One of them is behavioral competencies, which are close to the knowledge that can be used in the workplace [24]. Competence with a behavioral approach means that competence is formulated with a verb, behavior/performance, and the conditions of execution that allow its evidence through learning outcomes.

2.1.2. Curricular Levels

Curriculum design encompasses the macro, meso, and micro-curriculum levels. The macro-curriculum level considers curriculum planning at the country level. The meso-curriculum level considers planning at the institutional level, and the micro-curriculum level considers planning at the course level [5,25]. Figure 1 represents the relationship between these three levels.

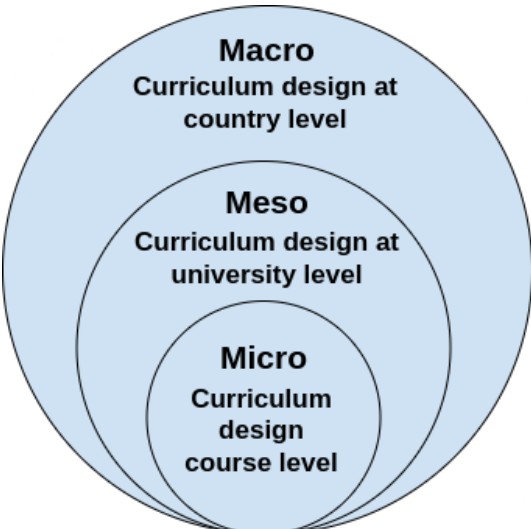

**Figure 1.** Curricular levels of curriculum design.

The macro-curriculum level corresponds to the education system in general; it includes minimum education and achievement indicators, among other general aspects [26]. It should outline the main lines of educational thinking, policies, and significant goals and integrates the highest curriculum design level. The principles and aims of this level are set out in the curriculum [27]. The principles and aims of this level are delineated by the State, represented in the administrations responsible for education. Legal provisions, resolutions, and laws regulate this level. This level includes didactic and assessment guidelines to guide teachers in their practice. However, it does not replace teachers in practice, in educational decision-making in schools and at the school level. It designates the decisions about what, when, and how to teach and assess. In general terms, at this level, plans and programs are developed at the level of the institution, for example, the Curriculum and Assessment Policy Statement of the Department of Basic Education in South Africa [28], or the basic curriculum for compulsory secondary education and the baccalaureate in Spain [29].

The meso-curriculum level is materialized in the educational institution's or intermediate bodies' project [27]. It enables the macro-curriculum to be specified in didactic proposals appropriate to its specific context and the aims and principles of the institution's management system [30]. This level articulates in the medium and long term the academic structure by areas and levels of training, according to the objectives, number of credits, and learning outcomes specified in the degree course syllabus. It decides what, when, and how to teach and assess. Generally speaking, it is at this level that the institution's plans and programs are developed. The meso-curriculum outlines the more general architecture of the training process that allows and facilitates micro-curricular structuring.

The third level, micro-curriculum, details the teaching–learning process through the didactic objectives, contents, development activities, assessment activities, and methodology of each subject that will materialize in the classroom, which is included in the analytical program and the corresponding study plan [27]. This level is the responsibility of each teacher and consists of planning the objectives, expected learning or learning outcomes, and didactic and assessment strategies for each group of students [26,30]. The teaching–learning process is detailed through the didactic objectives, contents, development activities, assess-

ment activities, and methodology of each subject that will materialize in the classroom, which is included in the analytical syllabus and the corresponding study plan.

In this sense, Soler considers the meso-curriculum as the discipline's design and defines micro-curricular design for the subject syllabus, topics or units, and classes [31]. According to the literature review on a micro-curricular design by Thomson [32], good practices are retrospective and iterative design approaches, constructive alignment, collaboration, distributed leadership, and feedback.

*2.2. Software Tools*

Software refers to computer programs, standards, rules, and documentation associated with an information processing system [33]. Software can be developed as a product for a specific customer or a general market.

Software development is becoming increasingly complex and must offer the correct performance and functionalities that users need. As such, the software must have quality aspects, such as those proposed by ISO 25010 [34], such as maintainability, reliability, and usability. Adopting a systematic and organized approach provided by software engineering allows developers to improve the quality of the developed tools [35,36]. This approach supports teamwork and meets the project development plan and the needs and expectations of stakeholders or the people who will use such tools. On the other hand, software engineering applies a systematic, disciplined, and quantifiable approach to software development, operation, and maintenance [37].

According to SWEBOK, the main activities of software development are software specification, software design, software construction, software evaluation, and software maintenance [38]. To this end, software engineering provides methods, models, and techniques that allow for improving software reliability and management of work teams and budgets [39]. The techniques allow for the improvement of recurring activities in development, formalizing a process that the entire development team must know. The methods allow for establishing the development process, the stages, and the products associated with each stage [25]. In addition, it allows for working with multidisciplinary teams and correcting errors as the plan progresses [40,41]. On the other hand, the models allow analyzing different points of view of the tool's design to be developed. Examples of these points are the data point of view, the users' point of view, the data flow, and technical aspects [42]. The following subsections detail important concepts about these aspects.

2.2.1. Models Used in Software Development

This section describes two models: (i) models used by the software development team for software analysis and design; and (ii) software quality models, which allow for improved relevance.

(i)  Software design models

Software developers use models that can come from different areas. On the one hand, software engineering models provide a notation and specific procedures for designing, analyzing, and developing models in software engineering [38,42]. One of the most widely used models is that proposed in the Unified Modeling Language (UML) since it provides a standard through rules and technical specifications for developers. UML considers two classifications: behavioral modeling (e.g., use case diagram, state machines, etc.) and structure modeling (e.g., class diagram, component diagram, deployment diagram, etc.). Figure 2 is an example of a class diagram, which shows the modeling of an online course. This figure shows relationships of association (Role–Account), aggregation (Course–Student), and composition (Student–Account). The cardinality of the relations can be 1..* (from one to many), 0..* (from zero to many), respectively.

On the other hand, other models do not belong to software engineering but to other disciplines. Of these models, ontologies and meta-models stand out, representing the entities and relationships of a specific domain or area. Ontology is a concept from philosophy representing aspects of reality [43]. An ontology in computer science is a model

that describes reality through concepts and relationships [44]. According to Stancin et al., an ontology is a specification of a conceptualization, which is used to identify relationships between concepts. Because of this, in education, ontologies are widely used in areas such as curriculum design and management, e-learning, and in the description of data and learning domains [45]. Figure 3 is an example of an ontology where the meaning of concepts and how they are related in the field of Learning Pathways in Higher Education is established [46].

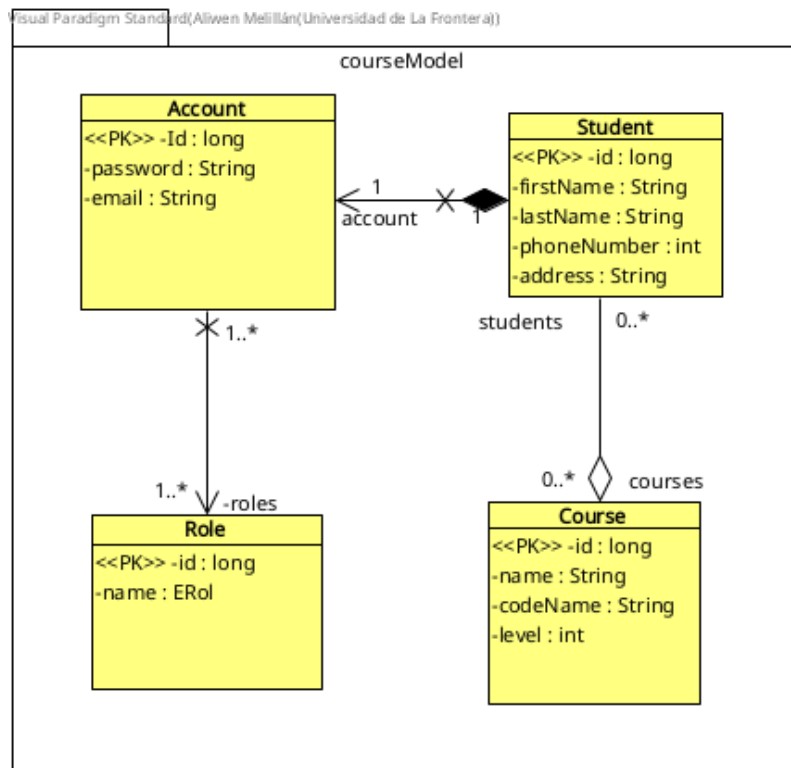

**Figure 2.** Class Diagram for Student and Course relationship.

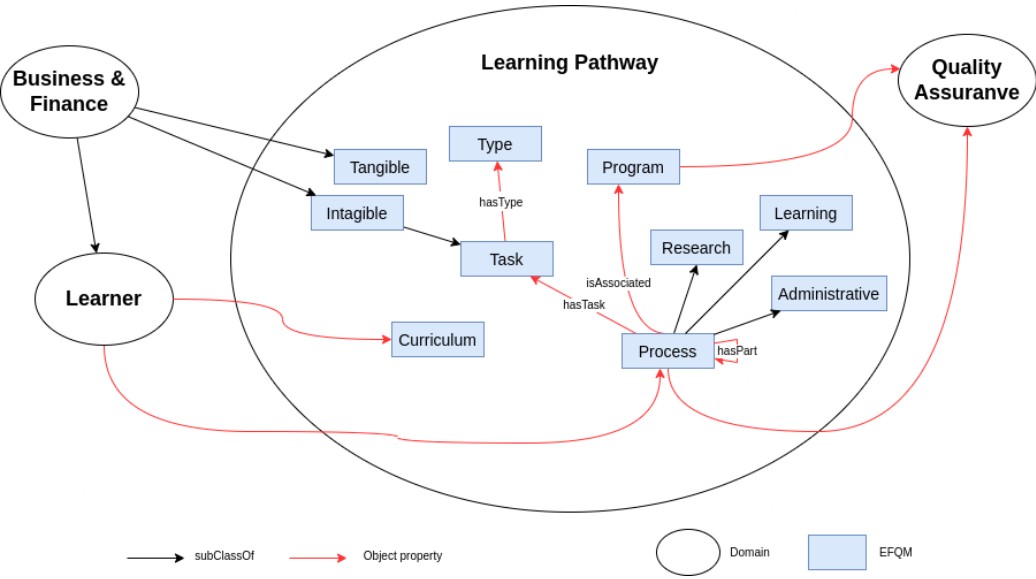

**Figure 3.** EDUC8 ontology model.

One model used in education is the IMS-Learning Design (IMS-LD) modeling language, which allows the design of learning units of a course [47]. IMS-LD, as a specification,

offers a conceptual model that defines the concepts and relationships in the instructional design domain. Its conceptual model is based on XML and is the successor of Educational Modeling Language. It considers concepts for a learning unit, such as the learning objective, prerequisites, roles, activities, and the environment where each activity will take place [48].

(ii)    Software quality models

Software quality models are models focused on ensuring that the product and process are developed professionally and efficiently. Callejas-Cuervo et al. evaluate the state of the art regarding these models, highlighting CMMI (Capability Maturity Model Integration), PSP (Personal Software Process), ITIL, COBIT 4.0, ISO25000, ISO9126, and Boehm, among others [49]. As an example of a quality model, CMMI presents five maturity levels in which the company or development team is placed based on criteria that must be met. Table 1 represents the CMMI maturity levels according to Chaudhary and Chopra [50]. One of the selected works uses CMM, a quality model prior to CMMI.

**Table 1.** CMMI maturity levels.

| Level | Name | Description |
|---|---|---|
| 1 | Initial | The process is uncontrolled and reactive, which makes it unpredictable. |
| 2 | Managed | The process is monitored, controlled, and reviewed but not common across the organization. |
| 3 | Defined | The process is proactive, documented, defined and common across the organization. |
| 4 | Quantitatively Managed | The process is controlled and measured. |
| 5 | Optimizing | The process is improved through continuous improvement. |

### 2.2.2. Software Development Methods

Software development methodologies, also called process models, provide a set of rules that define how to carry out a software development process, i.e., its activities, actions, tasks, the degree of iteration, the work products, and the organization of the work to be performed [51,52]. In this category, we can find traditional methodologies, such as the waterfall model and iterative, and new agile methodologies, such as Scrum or Extreme Programming (XP). There are also more specific methods, such as the Rational Unified Process (RUP), CREWS-SAVRE, or KAOS, in software requirements specification [53].

The waterfall model is one of the oldest software development methodologies. It consists of a systematic and linear process, contemplating requirement specifications, planning, modeling, construction, and software deployment [54].

The iterative or incremental method contemplates the same activities as the waterfall model but develops in minor versions (increments), where each version is an interaction of the previous one with improvements or new functionalities [39].

Agile methodologies attempt to cope with the changes that may occur in the development of a project by emphasizing the collaboration of people, their creations, and the speed to respond to change in processes, planning, and documentation [55]. Within the educational context, there is the ADDIE methodology. ADDIE (Analyze, Design, Develop, Implement, and Evaluate) is an instructional design approach that can be used as an educational software development methodology [56]. Applying this methodology makes it possible to respond to multiple learning contexts that allow an understanding of the requirements for educational software.

### 2.2.3. Software Development Techniques

Techniques in software engineering are considered as actions to improve some components of software development. Examples of software techniques are software testing [57], machine learning algorithms [58], continuous integration [59], or static code analysis for quality assessment [60].

### 3. Method

The method used is that of systematic mapping of studies, which is a methodology that allows establishing a conceptual map of knowledge in an area by focusing on more general research questions [61]. The protocol considered the guidelines proposed by Petersen et al. [19]. Figure 4 shows the systematic mapping process.

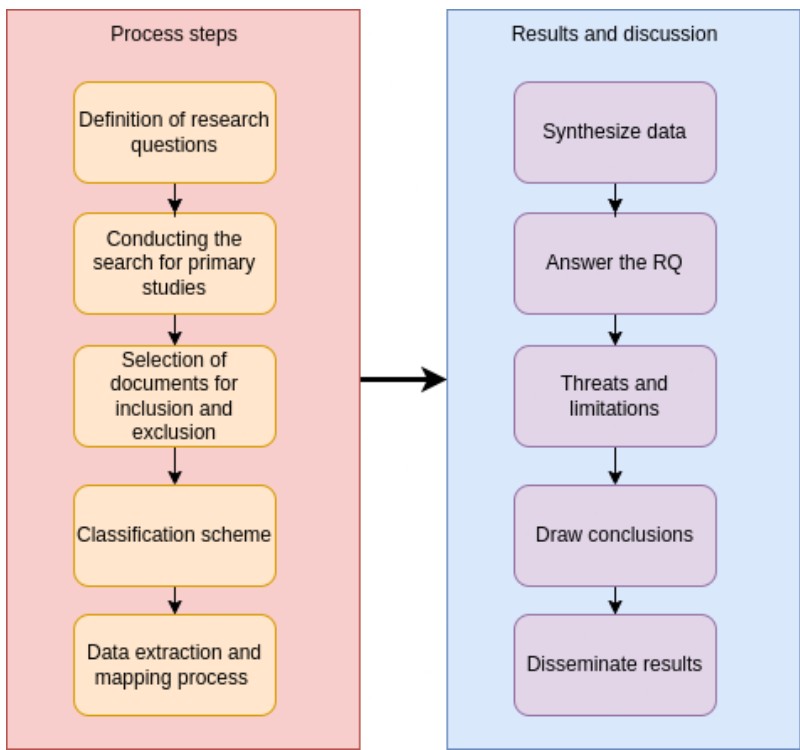

**Figure 4.** Systematic mapping process.

### 3.1. Stage 1: Definition of Research Questions

The design and implementation of student-centered curricula have gained importance due to the need to create skilled professionals for current industry needs [62]. Although the student-centered curriculum is essential in all areas of higher education, curriculum management, and therefore, assistive technology that allows managing the process from the point of view of the student, the teacher, and the team striving to achieve curricula that meet stakeholder expectations, has been considered critical [62]. Aligning the curriculum with market needs and the learning outcomes included in the subject plans is essential to meet these expectations [63]. Although software tools support this design process, information on how they are developed has yet to become available. This is important since the software must be aligned with the business objectives [64].

In this document, we have analyzed the support tools for the curriculum design process at the micro, meso, and macro-curricular levels to analyze the models that were used for the development of these software products since they allow identifying the curricular elements that were used and the relationships between them, showing the coherence between the software and the curriculum design obtained. At the micro-curricular level, it is expected to find models that identify elements such as learning outcomes, teaching–learning methods, and evaluations, among others. On the other hand, at the meso-curricular level, it is expected to find elements such as program profile, competencies, and result indicators. At the macro-curricular level, the elements are defined by each country.

On the other hand, the software development techniques and methods that were used were analyzed, allowing to identify aspects of rigor and quality of the tools used. It is expected to find methods currently used, such as agile methods. It is also expected to find

development methods that are unique to the area of education and therefore contribute to improving the alignment between IT and curriculum design.

The objectives to be developed in this systematic mapping are as follows:

O1.  Identify software support tools for curriculum design in education;
O2.  To analyze the use of techniques, methods, and models for the development of these tools;
O3.  Classify the curriculum stage solved by the software tool;
O4.  Identify the problems in curriculum design that are attempted to be solved with software tools.

These objectives give rise to the research questions (RQ) in Table 2.

**Table 2.** Research Questions.

| ID | Research Question | Justification |
|---|---|---|
| RQ1 | How has the development of curriculum design support tools evolved? | To identify the trends in the use of curriculum design support tools. |
| RQ2 | What is the curricular level that the technological tools support? | For classification at the macro-curricular (government), meso-curricular (career), or micro-curricular (course) level. |
| RQ3 | What models, methods, and techniques are used in developing the technology tools? | To identify whether the tools use models, methods, and techniques to enhance tool development. |
| RQ4 | What technological tools focus on solving the alignment between curricular elements? | To identify if it uses any model that allows for curriculum alignment or considers it in its study. |
| RQ5 | What problems motivated the development of the tool? | To identify the problem of why the tool was developed. |
| RQ6 | What challenges do the studies mention? | To identify remaining challenges or gaps that can be improved. |

### 3.2. Stage 2: Conducting the Search for Primary Studies

The search string incorporates key concepts related to technological tools, software, systems, and curriculum design. Curriculum design includes sub-processes of curriculum design, such as course design, instructional design, curriculum mapping, or curriculum management. Competency-based learning is also considered part of the new focus higher education institutions give to curriculum design. The data sources selected are Scopus, WoS, IEEE, and ERIC, as shown in Table 3.

**Table 3.** Data sources and search string.

| Source | Search String | # Articles |
|---|---|---|
| Scopus | KEY ("software" OR "tool" OR "system") AND KEY ("curricul* design" OR "curricul* mapping" OR "curricul* management" OR "course design" OR "instruction* design" OR "competency based learning") AND PUBYEAR > 2010 AND PUBYEAR < 2023 | 1608 |
| WoS | AK=("software" OR "system" OR tool") AND AK=("course design" OR "instructional design" OR "curricul* design" OR "curricul* mapping" OR "curricul* management") AND 2011-01-01 to 2022-12-31 (Publication Date) | 56 |
| IEEE | ("Index Terms": software" OR "Index Terms": tool" OR "Index Terms": "system") AND ("Index Terms": "curricul* design" OR "Index Terms": "curricul* mapping" OR "Index Terms": "curricul* management" OR "Index Terms": course design" OR "Index Terms": "instruction* design" OR Index Terms": "competency based learning") Filters Applied: 2011–2022 | 514 |
| ERIC | SU ("software" OR system" OR tool") AND SU ("course design" OR "instructional design" OR "curricul* design" OR "curricul* mapping" OR "curricul* management") publication date: 20110101-20221231 | 612 |

### 3.3. Stage 3: Selection of Documents for Inclusion and Exclusion

In order to filter the results, inclusion and exclusion criteria are applied. The inclusion and exclusion criteria are mixed according to the order in which they will be applied.



The inclusion criteria consider (C1) Articles in Spanish and English; (C3) Articles from journals, lectures, congresses, technical reports, theses, projects, and conferences; and (C5) Articles oriented to the use of a computer tool for curriculum design. Exclusion criteria considered: (C2) Articles prior to 2011 and after 2022; (C4) Articles as company web pages; (C6) Short articles of less than four pages; and (C7) Articles not available.

The search was performed for each database, and then data extraction was entered into a spreadsheet. The search yielded 2790 articles: 1608 in Scopus, 56 in WoS, 514 in IEEE, and 612 in ERIC (EBSCO). The information extracted from each article included the title, authors, keywords, abstract, year of publication, name of the publication source, and DOI number. From these data, 249 duplicate articles were identified and moved to another sheet in the database, resulting in 2541 articles as candidates for review.

Of the 2541 articles screened, 2464 were excluded due to the application of the exclusion criteria. The criteria applied were based on the review of each article's title, abstract, and keywords since they must incorporate the phrase "curriculum design" or "instructional design". In addition, articles whose title indicated the use of support tools at some curriculum design stages were included. This criterion was corroborated by reading the abstract. Once this criterion was applied, 77 relevant articles were obtained.

Of the 77 articles, two were excluded because they were unavailable in scientific databases. Seventy-five relevant articles were obtained, which should have been analyzed with another exclusion criterion. The criterion indicated that papers that contained less than four pages were a part or continuation of another article, and those that did not consider a tool or model should be eliminated. This left only 45 relevant articles.

In addition, recent reports on curriculum design processes were reviewed to search for studies that included support tools. However, none were found. Figure 5 shows the extraction process according to the PRISMA method with Haddaway et al., online tool [65].

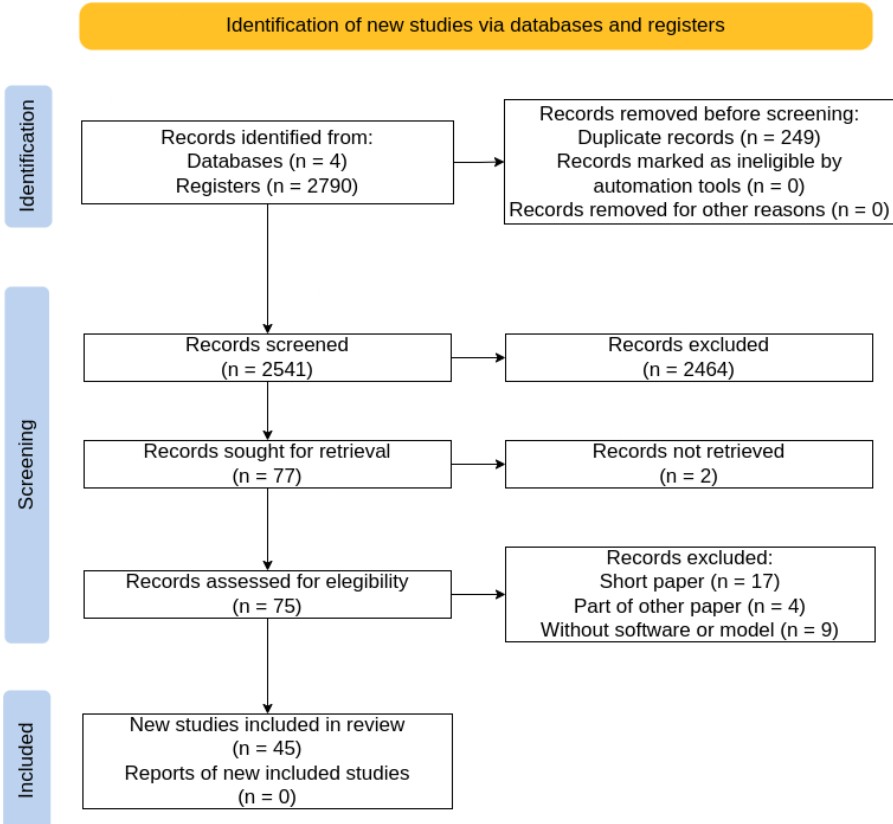

**Figure 5.** PRISMA flowchart.

*3.4. Stage 4: Classification Scheme*

Four of the six proposed RQs can be classified (RQ2, RQ3, RQ4, RQ5). Figure 6 shows the classification scheme for this study.

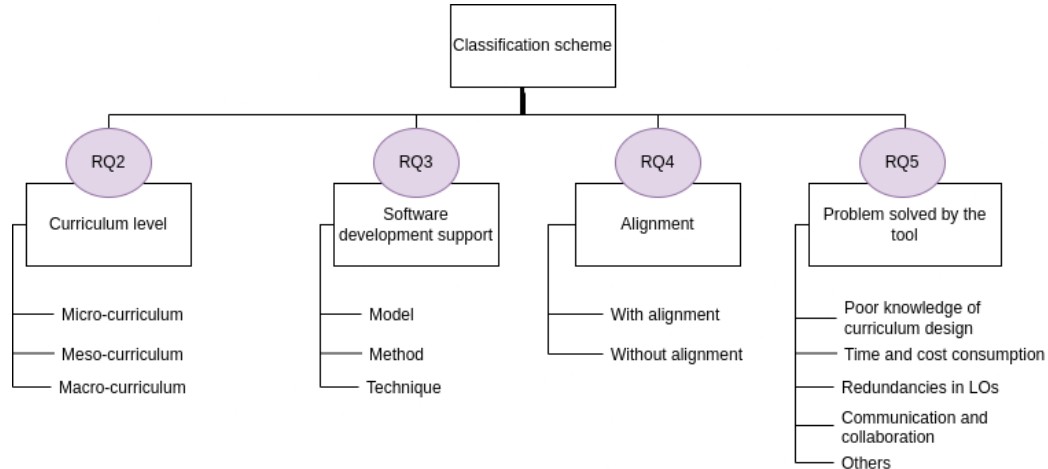

**Figure 6.** Classification scheme.

RQ2 classifies the curricular levels for which the tool is intended, which can be macro-curricular (country-level curricular planning), meso-curricular (institution-level planning), and micro-curricular (course-level planning), according to the classification of Chen-Quesada and Salas-Soto [5].

RQ3 classifies the tool according to some model, method, or technique. A model will be considered when curricular elements are represented through a scheme, which can be formal or not. Formal models include restrictions that make it possible to explain the links between these elements. On the other hand, a method corresponds to a sequence of steps, or a description of a process, to support curriculum design. The methods are used to develop the software, and finally, the techniques correspond to activities or tasks that improve the curriculum design. Software engineering models and methods are described in SWEBOK [38]. We have included the techniques since they allow us to identify good development practices [66].

RQ4 classifies the article according to whether it mentions curriculum alignment. This classification seeks to determine whether the reviewed paper mentions alignment as a problem to be solved. The alignment problem can be at the micro, meso or macro-curriculum level [67]. On the other hand, some papers mention the constructive alignment problem related to student learning [68].

RQ5 classifies the article according to the main problem solved by the software tool. These can be poor instructional design knowledge [69], time consumption [70], redundancy of learning outcomes [12], communication/collaboration [71,72], or others found in the selected articles.

*3.5. Stage 5: Data Extraction and Synthesis*

Data are extracted by reading each selected article. Only crucial data were extracted to answer the RQs. The following section describes the results obtained.

**4. Results**

Using the Cabuplot tool  [73], a categorical bubble chart is created for systematic mapping studies (Figure 7). The graph relates the use of models, methods, or techniques to the years in which the tools were published. The graph shows that models dominate the articles and their tools concerning other types during the periods covered by this systematic mapping (2011–2022).

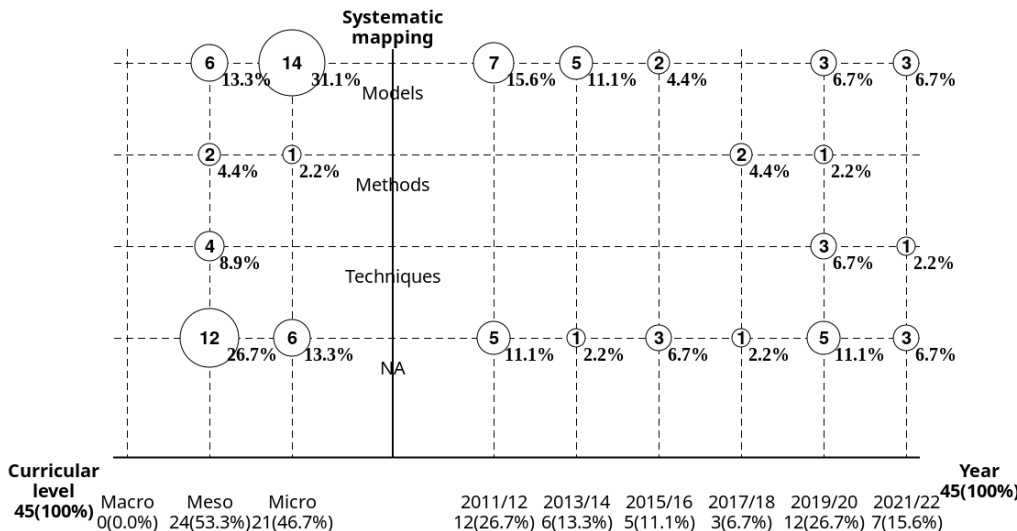

**Figure 7.** Categorical bubble chart for systematic mapping studies.

On the other hand, in terms of the year of publication of the articles, an increase in publications is visualized in recent years, which suggests that the problems of curriculum design are still current. In recent years, techniques and methods have begun to be used to develop software to support curriculum design, which shows progress in terms of formality in the development of applications.

As a result of this study, each research question (RQ) is answered. The details for each question are presented below.

## 4.1. RQ1: How Has the Use of Models, Methods, and Techniques Evolved

Figure 8 shows models, methods, and techniques for developing curriculum design support tools from 2011 to 2022. Software engineering models such as Unified Modeling Language (UML) and Boehm's model; IMS Learning Design (IMS-LD), used to model software in education; Capability Maturity Model (CMM), a quality model; and ontologies have been used. Ontologies are the most widely used models since they allow the representation of key concepts through a conceptual model. On the other hand, some methods have been in use since 2017. The methods mentioned by the authors are Rational Unified Process (RUP), Agile methodologies, prototypes, and ADDIE methodology. Only in the last years (2019–2022) have techniques been used in software development. In this case, they are related to using artificial intelligence, such as the Wordvec model and machine learning.

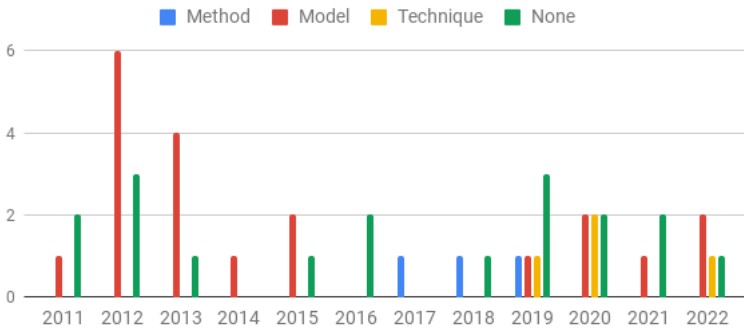

**Figure 8.** Graph of models, methods, or techniques for developing curriculum design support tools.

Figure 9 shows that ontology-based models have been used over time. Their use has increased in recent years. This is because it is a simple model that allows related elements,

allowing a better understanding of software development [74]. On the other hand, despite being an international standard, UML was only used in two articles during the years 2012 and 2013. Regarding methodologies, it is observed that ADDIE, which is used in education, has been used only twice. In addition, agile methodologies were mentioned only once, despite being the most used in the software market in recent years [75].

**Models, methods and techniques used**

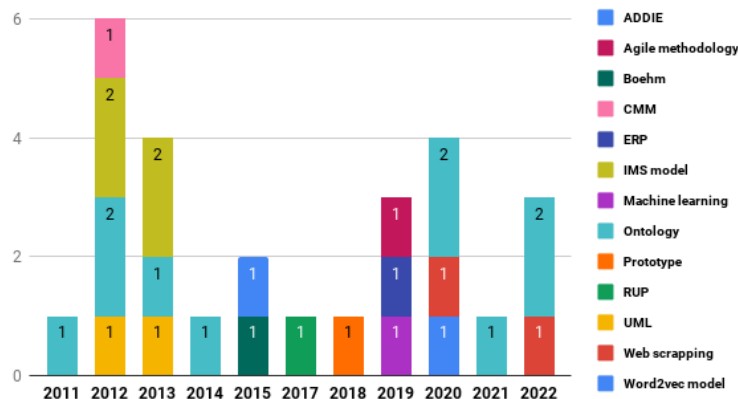

**Figure 9.** Models, methods, and techniques used.

*4.2. RQ2: What Is the Curriculum Level Supported by Technological Tools?*

The curricular levels supported by technological tools are the micro-curriculum level (21 tools) and meso-curriculum level (24 tools). However, there are meso-curriculum tools that also support micro-curriculum design. No article mentions the macro-curricular level, that is, any technical support related to curricular plans from any country's government.

*4.3. RQ3: What Models, Methods, or Techniques Are Used in Developing Technological Tools?*

From the 45 articles selected, 27 present some model, method, or technique in software development, described in Section 3.4. A summary is presented in Table 4. Most commonly used are ontology models to model the instructional process [76,77], course design [78,79], competencies [80–82], effective lessons [83], contents [84] or meso-curriculum components [85].

Chimalakonda and Nori present an ontology that explains the relationships between instructional design concepts such as learning objectives, teaching processes, and the didactic material used to reach the objectives [76]. Romero and Gutierrez present an ontology that represents the assessment activities according to the competency-based model so that teachers, curriculum experts, and other stakeholders can collaborate to improve the relationship between competency assessment and the means to achieve this assessment, such as teaching methods [80].

On the other hand, Sarkat and Negi use UML to visualize the design process and to model various components of an online course [86], while Jiang uses it for an instructional design system [87].

**Table 4.** Models, methods, or techniques used.

| Name | Category | Reference |
| --- | --- | --- |
| ADDIE | Method | [88] |
| Boehm | Model | [89] |
| CMM | Model | [90] |
| Prototype | Method | [91] |

**Table 4.** *Cont.*

| Name | Category | Reference |
|---|---|---|
| Entity Relationship Diagram | Model | [92] |
| IMS-LD | Model | [93,94] |
| IMS-QTI + IMS-RDCO | Model | [95] |
| IMS-LD + UML | Model | [96] |
| Machine learning | Technique | [97] |
| Agile methodology | Method | [32] |
| Ontology | Model | [76–85] |
| Rational Unified Process (RUP) | Method | [98] |
| UML | Model | [86,87] |
| Web Scraping | Technique | [99,100] |
| Word2vec model | Technique | [70] |

*4.4. RQ4: Which Technological Tools Focus on Providing a Solution to the Alignment between Curricular Elements?*

There are 21 articles that mention or attempt to provide a solution to the alignment problem. The alignment mentioned by the authors can be focused either at the meso-curriculum level or at the micro-curricular level as shown in Table 5. Regarding this matter, out of the 21 articles that mention alignment, 15 pertain to tools for the meso-curriculum level, while the remaining six relate to tools for the micro-curriculum level.

**Table 5.** Alignment mention by curriculum level.

| Curriculum Level | Reference |
|---|---|
| Meso-curricular level | [14,89–92,99–108] |
| Micro-curricular level | [14,32,79,80,86,90,95,101–103,109] |

At the meso-curriculum level, the focus, in general, is on aligning course objectives with the program [92,104,106,108] and with industry needs [89,99,100,107]. The micro-curricular level focuses on learning outcomes with assessment activities, learning methodologies, etc. [14,90,102,103]. In general, the issues that these tools attempt to solve are beyond the alignment of curricular elements. The solutions presented can range from software systems [14,91,95,102–104,107,108] to models based on ontologies, UML or quality models [79,80,86,89,90,92]. Arafeh seeks to improve micro-curriculum and meso-curriculum alignment and coherence through a course-level content mapping tool that helps better understand course content and products and plan and communicate a syllabus [101]. Dafoulas et al. [109] proposes tools for transforming curriculum design documents into an XML-based information model for course data, which utilizes a semantic similarity algorithm to align descriptions of different courses in a structured model. Meanwhile, Norcross et al. developed web scraping techniques to analyze job postings and compare them with curriculum design, aiming to identify curricular alignment between the current and future job market for upcoming graduates [99]. Karakolis et al. [100] also explored this concept through their system for aligning with the needs of the labor market. Florian-Gaviria et al. [95] offer several tools to align curricular components, such as learning outcomes, course requirements, learning activities, and assessment, with the European Qualifications Framework (EQF). In Slack, they developed tools using SQL queries and Python scripts to generate a course catalog document and prerequisite diagram [106]. Using their curriculum mapping tool, Oyewumi et al. aligned their medical curriculum with the topics of otorhinolaryngology and head and neck surgery [105].

*4.5. RQ5: What Problems Were Identified in the Study and Gave Rise to the Development of the Tool?*

Knowledge in instructional design is the problem that most attempts are made to solve. This problem encompasses a lack of quality due to a lack of instructional design knowledge [76,78,110] or being more researchers than teachers [111].

Curriculum management is a problem that encompasses the challenge of handling frequent changes [98] or a lack of formal method systematization for efficient management [89,102,108]. According to Bruno et al., developing digital learning materials for instructional design requires many human and economic resources [96]. Woo et al. mention that expert curriculum development is costly, time-consuming, and can be inconsistent due to subjective content selection by experts [70]. Thong et al. mention that it takes effort and time to design a curriculum without a computer system [90]. In Ling et al., they add that there is also a possibility of error [91]. Ball et al. mention the difficulty of designing a curriculum due to how changeable students' needs are and how influenced curriculum designers are by their experience and not by evidence. These difficulties can affect student performance [97]. Arafeh mentioned that good practices not always applied in curriculum development and implementation, such as discipline standards, industry requirements, and course alignment with content, activities, and assessments [101]. Oyewumi et al. created a tool to identify gaps and redundancies in a medical curriculum that is also time- and resource-consuming and has a variety of specialties that have to be coupled into it [105]. Matute et al. highlight weaknesses in curriculum design processes with little standardization [92]. Katsamani et al. attempt to bridge the gap between the design and implementation of an online course where teachers create Learning Designs (LD) in a narrative format without a standard template, making it difficult to disseminate and reuse [93]. The tools proposed by Norcross et al. [99], Tee et al. [107] and Karakolis et al. [100] focus on achieving alignment between curricula and industry needs, thereby helping students have realistic employment expectations. A summary of the issues identified is shown in Table 6.

**Table 6.** Main issues identified.

| Issues | Reference |
|---|---|
| Accreditation | [14] |
| Alignment with industry needs | [99,100,107] |
| Alignment of the course, the program, or the institution | [32,79,86,95,104,112] |
| Educational gaps, redundancies or lack of standardization | [85,92,93,97,101,105,109] |
| Teacher instructional design knowledge | [76–78,82,87,110,111,113,114] |
| Resource consumption (time and cost) | [70,90,91,96] |
| Designing courses efficiently | [83,84,88,94] |
| Curriculum management | [89,98,102,106,108,115] |
| Management of learning outcomes and/or competencies | [103,116,117] |
| Support to the activity evaluation process | [80,81] |

*4.6. RQ6: What Challenges Do the Studies Mention?*

Several challenges for software development to support curriculum design are mentioned in the selected articles. Another challenge is using various methodologies that must be adapted to a specific domain, for example, tools limited to instructional design patterns [76]. Technological tools for curriculum design are still facing quality challenges [86], which are reflected in their lack of usability due to a lack of consideration for the technology proficiency levels of teachers [83]. Therefore, authors such as Albo et al. are striving to improve the usability of their tools by addressing issues related to ease of use [114]. Other examples are prototypes or scripts that are not usable for a typical user [106]. For web scraping tools, developing these tools can take a long time since the extracted data have to be reviewed manually, leading to human error. Another challenge is the geographical and market limitation since the analysis of data from other regions is not possible due to the amount of data to be analyzed [99]. Furthermore, predicting that the data used in one region will give the same result as in other regions is impossible. Lai et al. mention that further studies on the implementation processes of online curriculum management tools would be beneficial [115]. Chimalakonda and Nori [76], and Sarkar and Negi [86] mention as possible future work the design of meta ontology or UML models, respectively. These guidelines suggest that the use of software engineering models is still in its infancy in curriculum design.

## 5. Discussion

This section includes an analysis and discussion from different perspectives. First, the use of models, methods, and techniques in developing curriculum design support applications is analyzed. Another perspective is the alignment problem since we are interested in looking for alternatives to improve the quality of the software and its practical use. Finally, we propose a model that represents the essential concepts of curriculum design, which should be addressed by the development team in order to improve alignment with the needs of stakeholders.

### 5.1. Use Models, Methods, and Techniques in Curriculum Design Support Tools

Regarding the use of models in software development, it was found that ontologies allow organizing the problem domain by integrating hierarchical and associative relationships between critical concepts [84].

On the other hand, ontologies allow the modeling of the objectives, the instructional process, and the instructional material to guide the development team [76]. Other authors used ontologies for the micro-curricular level so that the course has consistency with educational theories and practices [77,78,80,83,84]. In addition, ontologies allowed for obtaining and sharing a common vision of the domain of teaching and learning techniques for competency-based curriculum design [82]. Using ontologies generally benefits communication, interoperability and specification, reliability, and reusability in systems engineering [118]. Ontologies provide normative models that improve the need for communication, collaboration, consensus, and understanding of the curriculum design domain between different system actors, including educational stakeholders and software engineers, while maintaining consistency and freedom from ambiguities.

Regarding agile methodologies for developing a tool, the improvement in collaboration with end users, prioritization of the most critical features, and continuous delivery of an incremental product are highlighted [32].

While curriculum design support tools use models, methods, or techniques, there are also tools that do not mention some use. Dragon and Kimmich Mitchell developed a system with concept graphs to relate course content and learning objectives where the model is the system [112]. For example, what is a learning objective, and how could it connect to its parts and be modeled?

Regarding support tools at the meso-curricular level, the incipient use of methods, models, and software development techniques was found, considering that the components that integrate this level are more complex since it includes general aspects of a program of study, such as profile, competencies, and domains [14,103,116].

There are also other tools that, by their nature, did not use models, methods, or techniques, either because they are based on other software [110,117] or because they have a simple purpose, such as processing and displaying results [99,106].

Within software engineering, some models are considered standard and are now widely used. UML is a standard for object-oriented modeling notation, widely used for requirement gathering, analysis, and design of software applications [39]. One of the advantages is that it is managed by Object Management Group (OMG), a consortium that maintains technology standards internationally which software companies use. Some well-known standards are Model Driven Architecture (MDA), Business Process Model and Notation (BPMN), and Common Warehouse Meta-model (CWM). UML contains a meta-model that allows adaptations to other more specific contexts, such as Data Warehouses or Software Product Lines [119,120]. As a standard, UML is adopted in the software industry, so it is preferred to other models.

Considering the importance of standards in software engineering, there need to be more curricular design support tools identified. This weakness is also observed in developing other software tools in the educational domain. Ibarra-Corona and Escudero-Nahón conducted a meta-synthesis regarding the use of software engineering principles in developing educational platforms [18]. This study considers weakness in using these principles

because the leading software developers of these tools are the teachers themselves, so, understandably, the software does not have software engineering principles, or there are errors in the final product.

### 5.2. Alignment of Curriculum Design

In the area of curriculum design itself, alignment is considered the most critical problem, representing the link to the other problems. In this study, some works define alignment as the main problem.

According to Majerík et al., to ensure adequate education, a balanced curriculum with optimal learning objectives must be developed [108]. Similarly, Sarkar and Negi [86] suggest that courses should be coherent and consistent. Dragon and Kimmich Mitchell [112] needed to organize all courses cohesively to meet specific high-level learning objectives. Thus, they developed a tool linking courses with their materials and assessments. Thomson et al. needed to align course design at the institutional level [32]. Sarkar and Begi manage to relate assessment with curriculum design through UML [86]. On the other hand, Gluga et al. managed to align course objectives with the program [104].

Other authors do not describe the alignment problem and focus on problems such as AR or competency management [116,117], resource consumption in time and cost [70,96], curriculum management [98,115], etc. These problems could be lessened by improving alignment. Alignment implies having the curriculum organized and having its processes, ARs, competencies, etc., be consistent with the program's objective. By having an organization, less time and cost will be consumed for a curriculum redesign. Otherwise, the problems identified will persist or only be solved on a small scale. One of the factors for the need for alignment in curriculum design is due to the common understanding among stakeholders [32], causing, in turn, the needs of each one to be unaddressed [89]. In addition, some work makes explicit the need for improved communication, collaboration [32], and consensus [104,111] between education entities, software engineers [76] and industry-related entrepreneurs [109]. However, this need is obfuscated by problems that all stakeholders must possess, which is domain understanding [76], and the medium where they can participate and make their needs transparent [32,104,111].

In this sense, the use of models, methods, and techniques for software development allows for improving these aspects, giving a shared vision of the curriculum design domain [82] and allowing for managing, coordinating, guiding, assisting, and evaluating the curriculum [80,89,92].

On the other hand, the tools span the micro-curricular and meso-curriculum levels. Some tools were cataloged at the meso-curriculum level, encompassing the micro-curriculum level. At the micro-curriculum level, course design is considered through some models, especially competencies, and instructional design, which focuses on content. Several authors consider micro-curriculum design, but macro-curriculum design is not considered, as no tool covers it.

### 5.3. Conceptual Map

Figure 10 presents a conceptual map that summarizes the findings about the curriculum design support tools. This conceptual map relates the tools with the models, methods, and techniques used based on the systematic mapping developed. It also considers the curriculum design concepts and actors it supports.

As we mentioned in previous sections, two types of models that support curriculum design tools are found. In the first place are the quality models, which help the curriculum design process. Among these models are CMM [90] and Boehm's model [89]. In second place are the models for software design. The models used come from various areas, which help to represent the concepts and relationships sought to be modeled in the tool. Of these models for software design, we can find models coming from education, such as IMS-LD [93,94,96], IMS-QTI and IMS-RDCO [95]; models coming from philosophy, such as

ontologies [76–85]; and models coming from software engineering, informatic systems, and computer science, such as UML models [86,87,96] and entity–relationship diagrams [92].

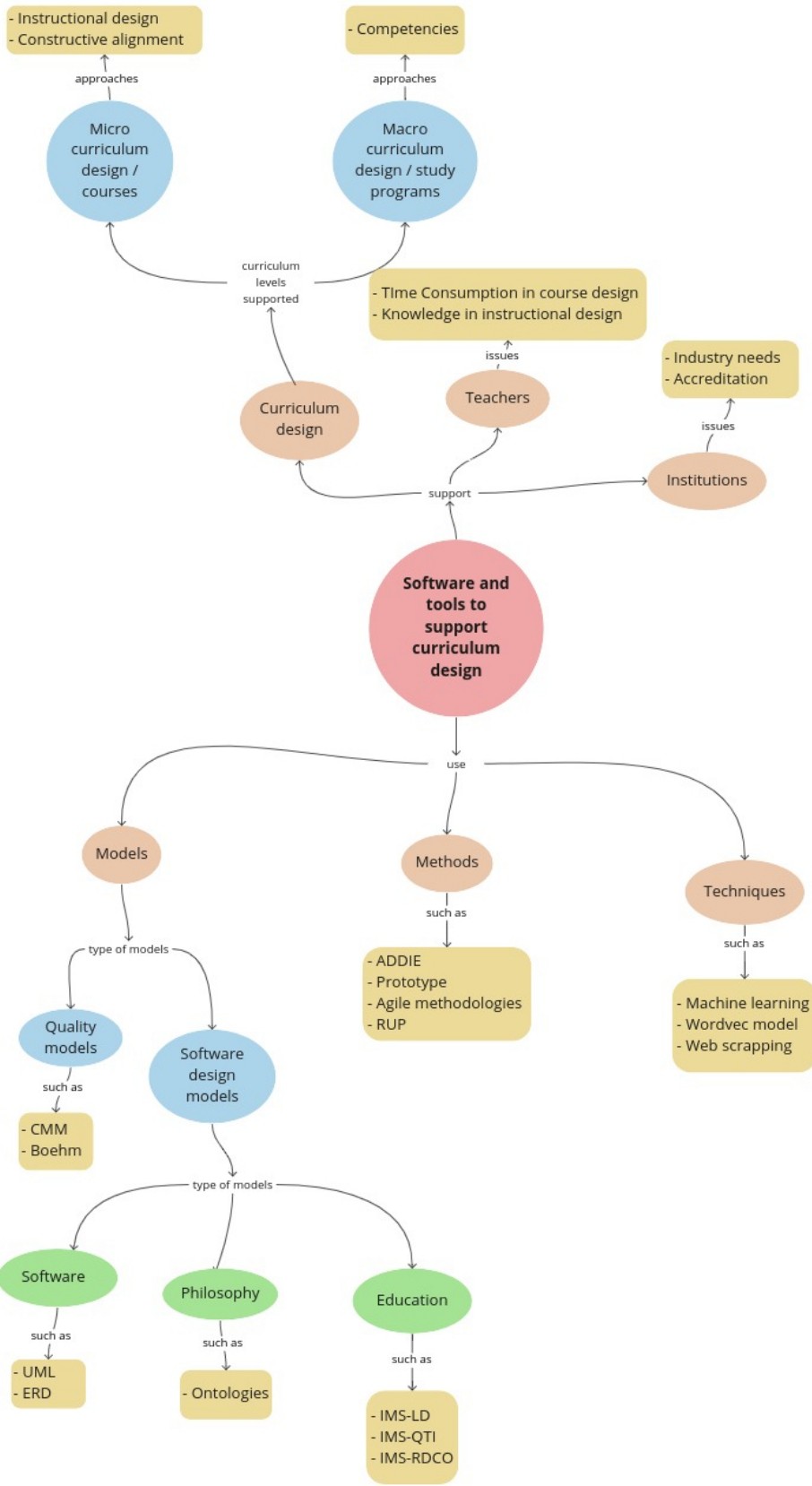

**Figure 10.** Concept map of tools to support curriculum design.

Software development methods used in the development of curriculum design support tools consider agile methodologies, development methodologies coming from education such as ADDIE [88], and traditional methodologies such as prototyping [91] and RUP [98]. Machine learning techniques [97], natural language processing [70], or web scraping [99] also support the development of tools to support curriculum design.

On the other hand, the tools found support certain concepts and actors found within curriculum design. In the first place is the institution. In this case, the technological tools provide ways to accredit their careers [14] and align the curriculum with the needs of the industry [99]. Secondly, there are the teachers, where technological tools help them to reduce the time they take to plan the courses they are involved in [70,90,91,96]. Likewise, technological tools help teachers who do not have mastery in curriculum design and/or instructional design [76–78,82,87,110,111,113].

Finally, the curriculum design support tools consider the domain of the micro and meso-curriculum levels. The micro-curricular design considers course design under instructional design and course design under constructive alignment. In the latter, learning outcomes management and competencies evaluation are highlighted [80]. Regarding the meso-curriculum design, it considers the concept of competencies in the context of the competency-based model [103,116,117].

Despite the advances in the development of tools to support curriculum design, studies have yet to be found that use meta-models to achieve communication and alignment between the technical software development team and the educational team. This issue is essential because meta-models provide the following benefits:

- A set of rules of association between the components of a system (e.g., curricular components);
- The definition of a common language among the professionals of the educational project;
- The generation of a working guide.

## 6. Related Work

Three relevant studies were found in the search for systematic literature reviews or study mappings. Table 7 shows the overlapping RQs for related work (✓: fully answered, ∼: partially answered).

**Table 7.** RQs and related work.

| Reference | RQ1 | RQ2 | RQ3 | RQ4 | RQ5 | RQ6 |
|:---------:|:---:|:---:|:---:|:---:|:---:|:---:|
| [18] | | | ✓ | | | |
| [14] | | | ✓ | | | |
| [2] | | | ∼ | | | |

Ibarra-Corona and Escudero-Nahón conducted a systematic literature review to analyze the principles of software engineering for developing educational technology platforms [18]. This study considered 69 articles from 2015 to 2020. The authors show the gap in the knowledge of the theory in educational technology, which causes a limitation in the quality qualities that the software should have. In addition, they present the use of software engineering and how modeling can help to identify pedagogical resources, analyze deficiencies and gaps, and evaluate quality. In addition, it is argued that software engineers should be involved in instructional design and teachers in the software development process. However, they recognize that both do not have training in the other area.

Carvajal-Ortiz and Florian-Gaviria conducted a systematic review of models and software applications to support the design, evaluation, and analysis of courses for competency-based curriculum management [14]. This study considered nine articles, from 2004 to 2019. The authors show that the software applications found have several aspects but do not consider all the characteristics supporting a competency-based curriculum. Likewise, the software tools consider different competency frameworks, conceptual hierarchies, and

evaluation methods. The authors then conclude that there is no single way to carry out curriculum design and competency assessment.

In Dodd [2], a study is conducted to provide an overview of curriculum design processes in various educational and professional contexts, highlighting the essential curriculum design competencies embedded in these processes. To this end, he analyzed various models of curriculum representation such as canvas, maps, and formats, which help to identify the elements at the micro and meso-curriculum levels. However, the author does not analyze tools to display and manage these models.

In this context, these related works are similar to our proposal as they are focused on the review of software for education support. They differ in which part of education they focus on. On the one hand, Ibarra-Corona and Escudero-Nahón focus on education in general [18]. On the other hand, Carvajal-Ortiz and Florian-Gaviria focus on software to support competency-based curriculum design. Our systematic mapping of studies focuses on curriculum design in general, considering competency-based curriculum design or instructional design [14]. Ibarra-Corona and Escudero-Nahón consider the use of software engineering principles in the development of software tools in education [18]. In addition, Dodd [2] only analyzes models for curriculum design and not for creating a tool or supporting software.

From the articles reviewed, no support tools for curriculum design are analyzed separately by curriculum levels. Furthermore, no information is available regarding the development of these tools, considering the software models used and the methods and techniques which allow analyzing the quality practices used. The models help to understand the curricular elements used and the relationships between them, defining clear rules for software development. On the other hand, the studies do not analyze the type of problem the tool solves. This is important to analyze the need versus the curricular elements involved. This gap allowed us to define the RQs for our study. Our systematic mapping of studies considers models, methods, and techniques used in software development in general, including software engineering principles.

## 7. Limitations

This section details the threats to the validity of the results and mitigation techniques, according to Petersen et al. [121].

### 7.1. Descriptive Validity

This validation describes whether the observations are objective and accurate. The associated mitigation measures were as follows.

- We used a Google spreadsheet, which stores and categorizes the RQs. This data collection form allows for uniformity and objectivity in the data and the data extraction process performed.

### 7.2. Theoretical Validity

This validation describes whether what is found is what was sought. During the paper selection process, some articles may not have been considered. The failure to not consider specific articles is due to three reasons: (1) the selection process was done at the individual interpretation of the first author, which presents the threat of subjectivity; (2) the choice of databases presents the threat of not obtaining all related articles; and (3) the search string considered only keywords, which may lead to fewer studies on curriculum design and its technological tool. The associated mitigation measures were as follows.

- In the case of the second point, the threat was mitigated through the use of four databases: one for education (ERIC), one for engineering and science (IEEE), and two covering all disciplines (Scopus and WoS).
- In the case of the third point, it is considered that the search string should consider various curriculum design concepts that should be part of the keywords of articles related to curriculum design support tools.

*7.3. Generalizability*

This validation describes the degree of generalizability of the results obtained. The associated mitigation measures were as follows.

- Various approaches to curriculum design, such as competency-based and instructional design, are considered.
- RQs are considered general enough to consider various curriculum design support tools at different curriculum levels and with various models, methods, or techniques used in developing the tool software.

*7.4. Interpretive Validity*

This validation describes whether the results are reasonable. The associated mitigation measures were as follows.

- The three authors perform a review and validation of the results.

*7.5. Repeatability*

This validation describes whether the inputs exist so that the study performed can be replicated by other authors. The associated mitigation measures were as follows.

- A complete sample of the systematic mapping process is shown in Section 3, which shows the search string used for each database and the inclusion and exclusion criteria chosen.

**8. Conclusions**

This paper presented a systematic mapping of studies to analyze literature regarding models, methods, and software development techniques in developing technological tools to support curriculum design. A set of 45 articles published between 2011 and 2022 was analyzed.

Curriculum design presents some problems, including teachers' poor knowledge of instructional design, which may lead them to make mistakes such as redundant learning outcomes or excessive time consumption to develop the curriculum. Another problem may be the communication or collaboration when teachers interact with software developers to design support software.

From our findings, we identified ontologies as the most commonly used model, while proprietary models and software engineering standards, such as UML, were very scarce. On the other hand, there is no evidence of tools that use meta-models to achieve communication and alignment between the software development technical team and the education team. Curricular alignment is one of the essential elements in the competency-based model for students to achieve the expected knowledge. One of the reasons for the lack of alignment is the lack of communication that the people involved can have so that they can mention their points of view and needs. Lack of alignment can then affect the management of learning outcomes and their relationship to teaching and assessment methods. It can also affect student learning. Models, methods, and techniques in software design and development can help software support curriculum design when the level of complexity is higher. Considering all the concepts and relationships of curriculum design, the use of models allows the communication and alignment of the concepts and relationships of the curriculum domain. These good practices allow for improved software quality.

We presented a conceptual map regarding software models, methods, and techniques and the elements they support within the curriculum design. This conceptual map gives a summarized view of findings in the systematic mapping and allows for generating a knowledge base for interdisciplinary work among curriculum specialists, program directors, teachers, and software developers.

In future work, we propose to develop a meta-model to solve the main problems encountered. Among these problems, we consider the alignment of curriculum design, curriculum management, management of learning outcomes, and support for the evaluation

process of activities. This meta-model would also support the teacher in understanding the concepts and relationships between the different curricular components.

**Author Contributions:** Conceptualization, A.M. and A.C.; methodology, A.M.; validation, A.M., A.C. and S.S.; formal analysis, A.M.; investigation, A.M.; resources, A.M.; data curation, A.M.; writing—original draft preparation, A.M.; writing—review and editing, A.C. and S.S. ; visualization, A.M.; supervision, A.C. and S.S.; project administration, A.C.; funding acquisition, A.C. All authors have read and agreed to the published version of the manuscript.

**Funding:** This research was funded by Universidad de La Frontera, Vicerrectoría de Investigación y Postgrado together with Vicerrectoría de Pregrado. Project IF22-0006.

**Institutional Review Board Statement:** Not applicable.

**Informed Consent Statement:** Not applicable.

**Data Availability Statement:** Not applicable.

**Acknowledgments:** Aliwen Melillán thanks Agencia Nacional de Investigación y Desarrollo de Chile (ANID)—Beca Magíster Nacional.

**Conflicts of Interest:** The authors declare no conflict of interest.

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
