# Peer review of "Software Development and Tool Support for Curriculum Design: A Systematic Mapping Study"

_applsci, doi:10.3390/app13137711_

Round 1
Reviewer 1 Report
I have carefully reviewed your paper and I am pleased to inform you that I recommend accepting it for publication with minor revisions. The paper addresses an important topic by analyzing software proposals for curriculum design support and exploring the incorporation of models, methods, and techniques from software development. I believe that this research will make a valuable contribution to the field.
Here are some specific comments and suggestions to further improve the paper:
-
Provide a clear and concise definition of curriculum design at the beginning of the introduction. This will help readers who may not be familiar with the topic to understand the context and importance of the study.
-
Discuss the rationale behind the selection of the six research questions. Why were these particular questions chosen? Were there any specific gaps in the existing literature that you aimed to address?
Here are some more specific comments/suggestions
1. Figure 2. Class diagram
The title of Figure 2 should be more specific. You should write an Example of Class Diagram for Student and Course relationship.
2. Figures 3, 5, and 6 are tiny. You should enlarge them
3. The use of models, methods, and techniques in technological tools to support curriculum design has been decreasing in recent years.
You need to explain why these models, methods, and techniques have been decreasing in recent years?
4. The authors need to explain the difference between method and technique. It seems that these two terms have different meanings in the paper (see Table 4).
Author Response
Dear Reviewer
Thank you for taking the time to comment on our work. Please find enclosed a letter in response to your queries, with explanations of the modifications and updates to the paper. We have highlighted in red color the modified sections and paragraphs.
Kind regards
Ania Cravero

Reviewer 2 Report
[Comment 1] Novelty
[Subcomment 1a] There are 2219 studies citing Petersen et al. [28]. The authors should have compared the novelty of their study with those previous studies (and the ones listed at the end of the manuscript) in a table, while comparing the different aspects, for clarity.
[Subcomment 1b] The contents of Section 6 must be provided in Section 1, close to the novelty statement.
[Comment 2] Review methodology and results
[Subcomment 2a] More details (e.g., the examples) must be presented in Figure 1, to clearly distinguish between the curriculum design at each curricular level. This explanation must be related to the analysis section.
[Subcomment 2b] (Figure 5) Please remove parts that have no studies (n=0).
[Subcomment 2c] (Figure 5) The authors must write reasons for excluding 2,464 studies in the screening stage.
[Subcomment 2d] (Figure 5) I do not understand why papers were excluded because of the following reasons (1) short paper and (2) part of other paper. These studies could include the exactly contents observed by the authors.
[Subcomment 2e] (Table 4) It would make sense for the authors to map the models, methods, and techniques based on their publication years and number of studies using it, to show which method is the best to be considered by the next researchers.
[Comment 3] Reference
Figure 6) Not all classifications were supported with any reference. The authors must cite references that explain the classifications.
[Comment 4] Writing quality and clarity
[Subcomment 4a] It is too difficult to understand the main points of the whole text. Please divide the explanations into several paragraphs based on their main idea.
[Subcomment 4b] The size of text in the figures must be as large as the main text to ensure readability.
Author Response

(The authors gave the same response as above.)

Round 2
Reviewer 2 Report
Thank you for your revisions.